# Complete Chloroplast Genomes and the Phylogenetic Analysis of Three Native Species of Paeoniaceae from the Sino-Himalayan Flora Subkingdom

**DOI:** 10.3390/ijms25010257

**Published:** 2023-12-23

**Authors:** Hanbing Cai, Rong Xu, Ping Tian, Mengjie Zhang, Ling Zhu, Tuo Yin, Hanyao Zhang, Xiaozhen Liu

**Affiliations:** Key Laboratory of Conservation and Utilization of Southwest Mountain Forest Resources, Ministry of Education, Southwest Forestry University, Kunming 650224, China; chb@swfu.edu.cn (H.C.); xurong@yxnu.edu.cn (R.X.); yiwangbaoyq@swfu.edu.cn (P.T.); zhangmengjie@swfu.edu.cn (M.Z.); zhuling@swfu.edu.cn (L.Z.); yintuo@swfu.edu.cn (T.Y.)

**Keywords:** Paeoniaceae, chloroplast genome, selection pressure, phylogeny, divergence time

## Abstract

*Paeonia delavayi* var. *lutea*, *Paeonia delavayi* var. *angustiloba*, and *Paeonia ludlowii* are Chinese endemics that belong to the Paeoniaceae family and have vital medicinal and ornamental value. It is often difficult to classify Paeoniaceae plants based on their morphological characteristics, and the limited genomic information has strongly hindered molecular evolution and phylogenetic studies of Paeoniaceae. In this study, we sequenced, assembled, and annotated the chloroplast genomes of *P. delavayi* var. *lutea*, *P. delavayi* var. *angustiloba*, and *P. ludlowii*. The chloroplast genomes of these strains were comparatively analyzed, and their phylogenetic relationships and divergence times were inferred. These three chloroplast genomes exhibited a typical quadripartite structure and were 152,687–152,759 bp in length. Each genome contains 126–132 genes, including 81–87 protein-coding genes, 37 transfer RNAs, and 8 ribosomal RNAs. In addition, the genomes had 61–64 SSRs, with mononucleotide repeats being the most abundant. The codon bias patterns of the three species tend to use codons ending in A/U. Six regions of high variability were identified (*psbK*-*psbL*, *trnG*-*UCC*, *petN*-*psbM*, *psbC*, *rps8*-*rpl14*, and *ycf1*) that can be used as DNA molecular markers for phylogenetic and taxonomic analysis. The Ka/Ks ratio indicates positive selection for the rps18 gene associated with self-replication. The phylogenetic analysis of 99 chloroplast genomes from Saxifragales clarified the phylogenetic relationships of Paeoniaceae and revealed that *P. delavayi* var. *lutea*, *P. delavayi* var. *angustiloba*, and *P. ludlowii* are monophyletic groups and sisters to *P. delavayi*. Divergence time estimation revealed two evolutionary divergences of Paeoniaceae species in the early Oligocene and Miocene. Afterward, they underwent rapid adaptive radiation from the Pliocene to the early Pleistocene when *P. delavayi* var. *lutea*, *P. delavayi* var. *angustiloba*, and *P. ludlowii* formed. The results of this study enrich the chloroplast genomic information of Paeoniaceae and reveal new insights into the phylogeny of Paeoniaceae.

## 1. Introduction

Paeoniaceae is a monophyletic family consisting of the genus *Paeonia* that is separated from Ranunculaceae and contains approximately 35 species. Stern (1946), in his magnum opus genus *Paeonia*, a study of the genus Paeonia, accepted Lynch’s division of the whole genus into three taxa, namely, the Sect. *Moutan*, the Sect. *Paeonia*, and Sect. *Onaepia* [1]. Among them, Sect. *Paeonia* and Sect. *Onaepia* are herbaceous plant types. Sect. *Paeonia* contains approximately 25 species distributed in Asia and Europe. Sect. *Onaepia* contains only two species distributed in southern North America. Sect. *Moutan* has eight species, which are woody types, all of which are endemic to China [2]. Kemularia-Natadze (1961) further proposed a five-group classification system [3]. However, some scholars believe that the carpels and indumentum of flowers should not be overemphasized, and a three-group classification system is more reasonable and natural. The results of chemical classification studies support the inclusion of the genus *Paeonia* as a separate family, as does the above morphological classification of paeoniflorin analogs at the group level; moreover, other plants in the *Ranunculaceae* do not contain paeoniflorin analogs. Sect. *Moutan* of the woody type has paeonol analogs, while Sect. *Paeonia* and Sect. *Onaepia* of the herbaceous type contains no one or very little [4,5,6]. Paeoniaceae plants have very high ornamental and medicinal value, and the roots and skins of most species can be used as drug agents. In the past hundred years, many scholars have conducted in-depth studies on the chemical composition of Paeoniaceae. Paeoniflorin is considered to be the main active compound in Sect. *Paeonia* root for relieving pain and has anticoagulant and anti-inflammatory properties [7,8]. However, dansylphenol is the main active ingredient in Sect. *Moutan* bark and is known to have anticoagulant, antioxidant, anti-inflammatory, and antiallergic properties [9]. In recent years, due to climate change and overexploitation, the populations of many Paeoniaceae have strongly declined [10,11], and many wild species of Sect. *Moutan* have been classified as near threatened or endangered by the International Union for Molecular Resources [12].

*P. delavayi* var. *lutea*, *P. delavayi* var. *angustiloba*, and *P. ludlowii* are endemic to the Sino-Himalayan flora subkingdom, and they are concentrated in the northwestern and central regions of Yunnan Province, as well as southeastern Tibet and southwestern Sichuan [13,14,15]. The taxonomic status of these plants has been disputed for a long time. The *P. delavayi* var. *lutea* and *P. delavayi* var. *angustiloba* are varieties of *P. delavayi*, and the main differences between them and the Dianthus peony are their flower colors and leaf widths. *P. delavayi* var. *lutea* flowers are yellow with purple patches at some bases. *P. delavayi* var. *angustiloba* leaves have narrow lobes, usually narrowly linear or lanceolate [16]. *P. ludlowii* was tall from a caespitose base and had relatively large, pure yellow flowers, yellow filaments, and typically one carpel per flower. Hong et al. elucidated the differences between *P. ludlowii* and *P. lutea*; subsequently, they separated “*ludlowii*” from *P. lutea* and treated it as a new species, identifying the existence of *P. delavayi* var. *lutea* and *P. delavayi* var. *angustiloba* as species [17,18]. The root bark of these plants is used medicinally to treat vomiting of blood, dysmenorrhea, and diabetes. Parts other than root bark can be used to treat pain in the chest, abdomen, and ribs; diarrhea; abdominal pain; and spontaneous and night sweats [19,20]. Moreover, *P. delavayi* and its variants are famous for their rich flower colors. This plant has rich genetic resources for determining flower color and is an indispensable breeding material for new peony cultivars. In recent years, rapid advancements in genomic technologies have strongly promoted research 79 on molecular breeding and functional genomics in species of Paeoniaceae [21].

Chloroplasts, as crucial plastids of plant cells, are organelles for photosynthesis and vital sites for molecule synthesis, such as pigments, starch, and fatty acids [22]. Chloroplasts have a relatively independent and complete genetic system compared with the nuclear genome, and the chloroplast genome sequence is highly conserved [23]. Angiosperms are generally maternally inherited, and gymnosperms are mostly patrilineally inherited [24]. The phenomenon of biparental inheritance is relatively rare, and it accounts for approximately 14% of angiosperms because it can provide more informative sites. The chloroplast genome plays a vital role in the plant phylogeny study [25]. The chloroplast genome is a closed-loop double-stranded DNA sequence that accounts for approximately 10–20% of the total DNA content of plants. A length of 120–220 kb is associated with two inverted repeats (IRs), which are 22–25 kb long and divide the whole chloroplast genome into a large single-copy region (LSC) and a short single-copy region (SSC) [26,27]. IR boundaries expand and contract with the evolutionary process of chloroplast genomes, which results in different degrees of sequence replication at the boundaries of each species. With the continuous development of sequencing technology, chloroplast genome sequencing methods tend to be simpler, and an increasing number of chloroplast genome sequences can be studied; these sequences are widely used in the fields of molecular breeding, phylogeny, conservation of endangered species, etc. [28,29].

In this study, we sequenced, assembled, and annotated the chloroplast genomes of *P. delavayi* var. *lutea*, *P. delavayi* var. *angustiloba*, and *P. ludlowii*. Comparative analyses of their chloroplast genomic features were performed to quantify their chloroplast genomic variation, and candidate chloroplast molecular markers were obtained. The phylogenetic relationships of Paeoniaceae have also been explored, laying the foundation for the study of the subgenus Paeoniaceae and its evolutionary history.

## 2. Results

### 2.1. Characterization and Comparative Analysis of Chloroplast Genomes

The chloroplast genomes of the three Paeoniaceae species sequenced in this paper were similar in size, with 152,759, 152,748, and 152,687 bp for *P. delavayi* var. *lutea*, *P. delavayi* var. *angustiloba*, and *P. ludlowii*, respectively. All of them conformed to the structure of the angiosperm chloroplast genome, which consists of four regions, i.e., two repetitive regions of the same sequence with the same orientation in the opposite direction, a large single-copy region, and a small single-copy region (Figure 1). The GC content ranged from 38.41 to 38.44, much lower than the AT content. The annotations yielded 132, 126, and 128 genes, respectively, containing 87, 81, and 83 protein-coding genes, 37 tRNAs, and 8 rRNAs, which showed that the differences in the number of genes in different species were mainly caused by protein-coding genes (Table 1). *P. delavayi* var. *lutea* has a deletion of the *trnR*-*UCU* gene, but it has eight tRNA genes (*trnI*-*CAU*, *trnL*-*CAA*, *trnV*-*GAC*, *trnI*-*GAU*, *trnA*-*UGC*, *trnR*-*ACG*, *trnN*-*GUU*, and *rps12*) that are duplicated. *P. delavayi* var. *angustiloba* and *P. ludlowii* have seven tRNA genes (*trnI*-*CAU*, *trnL*-*CAA*, *trnV*-*GAC*, *trnI*-*GAU*, *trnA*-*UGC*, *trnR*-*ACG*, and *trnN*-*GUU*) that were duplicated. In addition, the *rpl22*, *matK*, and *ycf15* genes were missing in *P. delavayi* var. *angustiloba*, and the *rpl22* and *psbL* genes were missing in *P. ludlowii*. Six tRNAs and ten protein-coding genes of the three Paeoniaceae species contained one intron, and *clpP* and *ycf3* had two introns (Table 2).

### 2.2. Identification of Long Repeats and Simple Sequence Repeats (SSRs)

Repeat sequences of longer length and complexity may strongly affect chloroplast genome rearrangement and sequence differentiation. Herein, we identified all repeats longer than 30 bp in the chloroplast genomes, and the results showed that the number of repetitive sequences in the chloroplast genomes of the three species of Paeoniaceae ranged from 40 to 45 (Figure 2). *P. delavayi* var. *angustiloba* and *P. ludlowii* contain forward, reverse, and palindromic repeats, while *P. delavayi* var. *lutea* contains only forward and palindromic repeats. Among them, 19–22 were forward repeats, 19–23 were palindromic repeats, and reverse repeats were detected in only one of the *P. delavayi* var. *angustiloba* and *P. ludlowii* accessions. Most of the repeat sequences were in the range of 30–39 bp.

The SSR identification of the chloroplast genome revealed that *P. delavayi* var. *lutea* and *P. delavayi* var. *angustiloba* contained 61 SSRs, while *P. ludlowii* had 64 SSRs. The percentage of mononucleotide repeats was the highest. In the three Paeoniaceae species, 40–43 were mononucleotide repeats, 12–13 were dinucleotide repeats, 5 were trinucleotide repeats, and 6 were tetranucleotide repeats (Figure 3A). Most of the mononucleotide and dinucleotide repeats were A/T and AT/TA, and most of the other types of repeats were also A and T as the basic repeating units, with C and G occurring very infrequently. Among the four chloroplast partitions, the highest content of SSRs was found in the LSC region (48–53), followed by the SSC region (8–9), and the lowest was found in the IR region (4) (Figure 3C). Most of the SSRs were simple, and compound SSRs were rare (Figure 3B).

### 2.3. Analysis of IR Boundary Changes

We compared the IR boundaries and locations of neighboring genes in ten closely related Paeoniaceae species (Figure 4). The LSC/IRb boundary was located at *rps*19 in all the plants except for *P. rockii*, where the gene was located at *rpl2*, which was 2–4 bp in length in the IRB region, and the *rpl2* gene in the LSC region was fully amplified to the IRb region. For the IRb/SSC boundary, both have a 2 bp gap with the *ndhF* gene. The SSC/Ira region was the most conserved region, and the SSC/Ira boundaries of all the species were located within the pseudogene *ycf1*, which ranges from 5417 to 5432 bp in length in the Ira sequence. Except for the Ira/LSC boundary of *P. rockii*, which was located at *rpl2*, the Ira/SSC boundary in all the other species was within the *trnH* gene, and the *trnH* gene was differentially extended to the Ira region, with a length of 1–6 bp in the Ira region.

### 2.4. Codon Usage Bias

In this study, the relative probability of codon usage was analyzed for all protein-coding genes in the chloroplast genomes of three species of Paeoniaceae, and the synonymous codon usage rates were calculated. The results showed that *P. delavayi* var. *lutea*, *P. delavayi* var. *angustiloba*, and *P. ludlowii* had 19,778, 19,901, and 20,456 codons, respectively, containing 64 codons and encoding 20 amino acids (Figure 5). Among all the amino acids, leucine (Leu) had the highest percentage of codons (10.21–10.36%), and cysteine (Cys) had the lowest (1.10–1.12%). The codon AUU, encoding isoleucine (Ile), was the most common codon, with 823–856 occurrences, and the codon UGCon, encoding cysteine (Cys), was the least common, with only 60–61 occurrences. This codon is the best, with a high preference if the RSCU > 1. There were 32 codons with usage preferences in the chloroplast genomes of all three species of Paeoniaceae. Twenty-eight of these thirty-two codons end in A/U bases. This finding suggested that amino acid codons prefer to end at the A and U bases in their chloroplast genomes. The codons with the highest and lowest RSCU values were UUA and GAC, which encode leucine and aspartic acid, respectively (Figure 5, Appendix A).

### 2.5. Sequence Divergence and Mutation Hotspot Regions

Using the chloroplast genome of *P. delavayi* as a reference, the sequence alignment analysis of the chloroplast genomes of *P. delavayi* var. *lutea*, *P. delavayi* var. *angustiloba*, and *P. ludlowii* revealed that the sequences were aligned in the same order and highly conserved. Compared with those of protein-coding regions, the noncoding regions were more different. Of the four chloroplast components, the LSC region had the highest variation, and the IR region had the lowest change (Figure 6).

The nucleotide diversity (Pi) of the chloroplast sequences was also analyzed in this study using DnaSP. The degree of variation in the chloroplast IR region was significantly lower than that in the LSC and SSC regions, with Pi values less than 0.005, which is consistent with the results of mVISTA. Five areas of nucleotide diversity were more prominent, and four were located in the LSC region, one in the SSC region, three in the intergenic region, and three in the gene region, with Pi values higher than 0.007. These highly mutated regions were located in *psbK-psbI*, *trnG-UCC*, *petN-psbM*, *psbC*, *rps8-rpl14*, and *ycf1* and contained 10, 9, 9, 9, 9, and 8 mutation sites, respectively (Figure 7).

### 2.6. Selection Pressure Analysis

The MLWL model was used to calculate the ratios (Ka/Ks) of synonymous (Ka) and nonsynonymous (Ks) substitutions for the protein-coding genes of *P. delavayi* var. *lutea*, *P. delavayi* var. *angustiloba* and *P. ludlowii*. The Ka/Ks values of the protein-coding genes of these three plants were similar, ranging from 0 to 1.1172, and most of the genes had Ka/Ks values less than 1. There were five genes with Ka/Ks ratios of 0, e.g., *petN*, *psaC*, *psbA*, *psbE*, and *rpl36*. Only the *rps18* gene, a small-subunit gene of ribosomal proteins, had a Ka/Ks greater than 1 (Figure 8).

### 2.7. Phylogenetic Analyses

To understand the phylogenetic position and evolutionary relationships of Paeoniaceae, the whole chloroplast genome (WCG) datasets of 99 Saxifragales species were aligned, and the maximum likelihood (ML) method was applied to construct the phylogenetic tree (Figure 9). The results showed that the Paeoniaceae were clustered into a single branch with the Daphniphyllaceae, Altingiaceae, Cercidiphyllaceae, and Hamamelidaceae and had relatively close affinities. The viewpoint in favor of affiliating Paeoniaceae with Saxifragalesis was supported.

Eighteen Paeoniaceae species and two outgroup species were aligned based on whole chloroplast genome (WCG) and protein-coding gene (PCG) datasets. Maximum likelihood (ML) and Bayesian (BI) analyses were applied to construct a phylogenetic tree (Figure 10). The phylogenetic tree showed inconsistent topology, the labeling ML tree’s bootstrap (BS) values, the BI tree’s posterior probability (PP) at the nodes, and the WCG tree showed high support, with all node posterior probability values > 0.95. Based on the evolutionary tree, 18 species in the Paeoniaceae family were divided into two large evolutionary branches, and the Sect. *Paeonia* species were distributed on different evolutionary branches from Sect. *Moutan* species, consistent with the results of chemical classification. Unexpectedly, the herbaceous *P. brownii* was more closely related to the woody Sect. *Moutan*. The differences between the WCG and PCG trees are described in Sect. *Moutan*. *P. ludlowii*, *P. delavayi* var. *lutea*, *P. delavayi* var. *angustiloba*, and *P. delavayi* formed a monophyletic group. However, in the PCG tree, they are a concurrent paraphyletic group, with *P. Ostii* peony and *P. delavayi* clustered into a single branch. The morphological classification results were consistent with the whole-gene tree, and the ovaries of *P. ludlowii*, *P. delavayi* var. *lutea*, *P. delavayi* var. *angustiloba*, and *P. delavayi* were always without membrane envelopes. The other six species on this evolutionary branch have a membrane encompassing the exterior of the ovary before it expands.

### 2.8. Divergence Time Dating

The divergence time of Paeoniaceae was obtained based on whole chloroplast genome data, and the mean estimated ages and 95% HPD intervals of nodes were mapped onto the dating chronogram (Figure 11). The crown age of Paeoniaceae is estimated to be 31.21 Ma (95% HPD: 29.05–34.55 Ma) in the Oligocene at the end of the Paleogene. The stem age of Sect. *Paeonia* and Sect. *Moutan* is approximately 21.19 Ma (95% HPD: 19.16–23.72 Ma) in the Neogene as well as in the early Miocene; their crown ages are 12.94 Ma (95% HPD: 11.59–14.56 Ma) and 13.49 Ma (95% HPD: 12.11–15.28 Ma), respectively, which are also from the Miocene. From the late Miocene to the Pleistocene, a dramatic diversification of Paeoniaceae species occurred. The common ancestor of *P. delavayi* var. *lutea*, *P. delavayi* var. *Angustiloba*, and *P. ludlowii* converged to *P. delavayi* during this period, with a stem age of 6.57 Ma (95% HPD: 5.74–7.51 Ma) and a crown age of 4.99 Ma (95% HPD: 4.33–5.74 Ma). *P. delavayi* var. *lutea* and *P. ludlowii* diverged at 4.58 Ma (95% HPD: 3.96–5.30 Ma).

## 3. Discussion

We sequenced *P. delavayi* var. *lutea*, *P. delavayi* var. *angustiloba*, and *P. ludlowii* using second-generation high-throughput sequencing technology (Illumina NovaSeq 6000 platform) in this study. We obtained whole-chloroplast genomes via assembly and annotation. The chloroplast genomes of the three species were identical to those of most angiosperms [30,31,32] and were divided into four parts, namely, a large, a small single-copy region, and two inverted repeats, with lengths of 152,687–152,759 bp. The GC content was 37.4, similar to that of other Paeoniaceae [33]. The total number of chloroplast genes ranged from 126 to 132, with differences mainly originating from protein-coding genes, with rRNAs being the most conserved, all with eight genes. Gene loss frequently occurs during the evolution of the cp genome in angiosperms, and contraction/amplification of IR regions can lead to gene loss/addition. According to the chloroplast genomes examined, the *infA* and *trnR-UCU* genes were missing in *P. delavayi* var. *lutea*; the *infA*, *rpl22*, *matK*, and *ycf15* genes in *P. delavayi* var. *angustiloba*; and the *infA*, *rpl22*, and *psbL* genes in *P. ludlowii*. *infA* is a transcription initiation factor, and this gene has been repeatedly reported to be translocated to the nucleus [34].

Vascular plant chloroplast genomes usually exhibit the contraction or expansion of boundaries within different genera or even within the same genus, which is the main factor leading to variations in the length and number of genes in various species [35,36,37]. The LSC/IRb boundary was generally located on the *rps19* gene in nine Paeoniaceae species, except for *P. rockii*, where the expansion of the IR region resulted in the *rpl2* gene being inside the IRb. The trnH genes were shifted to varying degrees to the IRa region, and expansion of the IR also resulted in partially replicated *ycf1* genes being in the IRb region, thus generating the pseudogene *ycf1* at the IRb/SSC boundary. Most land plants always have pseudogenes in their chloroplast genomes at the SSC/IRa boundary [36,37].

Repetitive sequences affect the transcriptional regulation of genes, protein translation, chromosome formation, and metabolism and can reflect differences between mutation frequencies and evolutionary rates of species [38]. The number of tandem repeats in the chloroplast genomes of the three Paeoniaceae species ranged from 40 to 45, with *P. delavayi* var. *angustiloba* and *P. ludlowii* containing forward, palindromic and reverse repeats and *P. delavayi* var. *lutea* containing only forward and palindromic repeats. Among all the repeats, those with lengths of 30–39 bp accounted for the highest proportion and were mainly forward repeats. SSRs are widely distributed in the genomes of organisms and can cause polymorphisms to form due to differences in the number of repetitive units [39]. Compared with the nuclear genome, the chloroplast genome is relatively small. However, there are also a certain number of SSRs that have the characteristics of stable heredity, high abundance, and wide distribution. It is widely used for determining genetic diversity and species identification [40]. Sixty-one to sixty-four SSR loci were identified in this study in Paeoniaceae; these loci contained four types of repeats. Mononucleotide and dinucleotide repeats were the most common, accounting for more than 80% of the repeats. These SSRs were distributed mainly within the LSC region, and most of them exhibited strong A/T preferences, with A and T bases serving as the basic repeats, which was attributed to the enrichment of polyamines and polythymines in the chloroplast genome.

Whole-genome sequences of *P. delavayi* var. *lutea*, *P. delavayi* var. *Angustiloba*, and *P. ludlowii* were compared with that of *P. delavayi* as a reference, and the results showed a high degree of conservatism in this study. The degree of variation was higher in the LSC and SSC regions than in the IR region, and the degree of variation was higher in the noncoding region than in the coding region. Sliding window analysis revealed several regions with a high degree of variation, such as *psbK-psbI*, *petN-psbM*, *psbC*, *rps8-rpl14*, and *ycf1*, which were primarily located in intergenic regions; among these regions, *ycf1* was the gene with the highest degree of variation (Pi: 0.78), and there were more variant sites. Boundary change analysis revealed that this gene is located at the boundary of SSC and IRa, and this gene can be considered a vital barcode for the subsequent identification of Paeoniaceae species.

The Ka/Ks ratio indicates the selection pressure acting on the gene, with the gene being subjected to positive selection when Ka/Ks > 1, neutral evolution when Ka/Ks = 1, and purifying selection when Ka/Ks < 1 [41,42]. Ka/Ks = 1 can be calculated for a gene not under natural selection pressure. However, this ratio is usually less than one because nonsynonymous substitutions usually result in deleterious traits and rarely result in an evolutionary advantage. The mean Ka/Ks values for the protein-coding genes of *P. delavayi* var. *lutea*, *P. delavayi* var. *angustiloba*, and *P. ludlowii* were 0.1909, 0.1920, and 0.1851, respectively, with 73 protein-coding genes having Ka/Ks values less than 1, suggesting that these genes were subjected to purifying selection. Only the *rps18* gene, associated with self-replication, was found in all three species. The self-replication-related *rps18* gene had a Ka/Ks greater than 1, indicating that this gene was affected by positive selection and was in a period of rapid evolution.

The taxonomic status of Paeoniaceae has been controversial, and some of the traits used for morphological classification are susceptible to environmental and geographic factors; these traits are variable, and it is difficult to determine their ancestry [43,44]. The chloroplast genome is highly suitable for phylogenetic studies due to its high conservation. Wu et al. compared Paeoniaceae with Ranunculaceae, and molecular evidence suggests that they are not closely related to Ranunculaceae [45]. This study supports the placement of Paeoniaceae in the order Saxifragales and revealed that Paeoniaceae has closer affinities with Daphniphyllaceae, Altingiaceae, Cercidiphyllaceae, and Hamamelidaceae. Within the Paeoniaceae, the species of Sect. *Paeonia* and Sect. *Moutan* are in two large evolutionary branches. *P. delavayi* var. *lutea* and *P. delavayi* var. *angustiloba* were recorded as *P. delavayi* in the Flora of China, and the traits differed from those of *P. delavayi* mainly in terms of the color of the flowers and the width of the leaf lobes [16]. However, they did not form a monophyletic group in the WCG tree but were interspersed with other species. *P. delavayi* var. *angustiloba* and *P. delavayi* var. *lutea* showed closer affinity to *P. ludlowii*. According to the WCG tree, *P. delavayi*, *P. delavayi* var. *lutea*, *P. delavayi* var. *angustiloba*, and *P. ludlowii* formed a monophyletic group. However, they were a paraphyletic group in the PCG tree. Among them, *P. delavayi* var. *lutea*, *P. delavayi* var. *Angustiloba*, and *P. ludlowii* formed a monophyletic group of sisters of *P. delavayi*. In addition, we found that *P. brownii* is more closely related to Sect. *Moutan* than to Sect. *Paeonia*. *P. brownii* is geographically distributed only in western North America, with an interrupted distribution compared with the other two groups. This distribution pattern is similar to that of *Liriodendron* (*Liriodendron chinense* and *Liriodendron tulipifera*) [46]. Moreover, flowers of *P. brownii* usually have several flowers per shoot, and flowers of European Paeoniaceae species are solitary, so it is unlikely that *P. brownii* diverged from European Paeoniaceae species. Combining divergence time and phylogenetic trees, we infer that the initial divergence of the Paeoniaceae occurred between Sect. *Onaepia* and Sect. *Moutan*.

Divergence time estimation revealed a crown age of 31.21 Ma (95% HPD: 29.05–34.55 Ma) for the Paeoniaceae in the Oligocene at the end of the Paleogene. Then, evolutionary divergence occurred in the Miocene, with a divergence time of 21.19 Ma (95% HPD: 19.16–23.72 Ma), which is the stem age of Sect. *Paeonia* and Sect. *Moutan*. The Paeoniaceae underwent dramatic diversification from the late Miocene to the Pleistocene, the closest period of stable and persistent warmth, with atmospheric carbon dioxide levels similar to those observed today. The significant enhancement of the Asian–African summer wind circulation leads to increased precipitation in Asia [47]. The common ancestor of *P. delavayi* var. *lutea*, *P. delavayi* var. *Angustiloba*, and *P. ludlowii* converged to *P. delavayi* during this period. This monophyletic group has a stem age of 6.57 Ma (95% HPD: 5.74–7.51 Ma) and a crown age of 4.99 Ma (95% HPD: 4.33–5.74 Ma).

## 4. Materials and Methods

### 4.1. Plant Materials, DNA Extraction from Samples, and Sequencing

*P. delavayi* var. *lutea* and *P. delavayi* var. *angustiloba* were collected from Lijiang, Yunnan Province, and *P. ludlowii* was collected from Linzhi, Tibet. The fresh leaves were removed, dried, and preserved in silica gel. The specimens were deposited in the Southwest Forestry University Herbarium. The total DNA of the plants was extracted by a modified CTAB method, the integrity of the DNA was detected by 1% agarose gel electrophoresis, and the concentration of the DNA was checked by a Qubit fluorescence photometer [48]. The quality of the DNA was decreased to 300–500 bp with a Covaris M220-focused ultrasonicator (Covaris, Woburn, Massachusetts, USA). Fragment purification, end repair, A-tailing of the 3′ ends, and ligation of the index adapter were performed to generate sequencing libraries. The libraries were sequenced on the Illumina NovaSeq 6000 platform (Illumina, San Diego, California, USA; BioLinker Biotechnology Ltd.). The raw data were then filtered using Trimmomatic-0.39, and the steps were as follows. The exclusion criteria for trimming were as follows: (1) no AGCT at the 5′ end; (2) trimmed reads with a sequencing quality less than Q20; (3) removed reads with a quantity of N higher than 10%; and (4) had the adapter and small fragments less than 75 bp in length removed after trimming. The sequences were filtered to obtain clean data of 7.06, 5.01, and 5.12 Gb.

### 4.2. Chloroplast Genome Assembly and Annotation

Chloroplast genome assembly was performed by GetOrganelle (v 1.7.7.0) [49]. The GetOrganelle first filtered plastid-like reads, conducted the de novo assembly, purified the assembly, and finally generated the complete cp genomes. The chloroplast genome coverage was 157.2×, 160.3×, and 150.9× for *P. delavayi* var. *lutea*, *P. delavayi* var. *angustiloba*, and *P. ludlowii*, respectively. The k-mer values are set to 65, 105, and 127; other commands use default parameters. The whole chloroplast genome sequence of *P. delavayi* (MN463100.1) was subsequently used as a reference and annotated with CPGAVAS 2 [50] and manually corrected using Geneious (v9.0.2). The raw sequencing data (Submission ID: 2717099, 2717123, 2717127) and the assembled chloroplast genome information (GenBank accession numbers: OR187760, OR187761, OR187762) were uploaded to the NCBI database.

### 4.3. Statistical Analysis of Characteristics of the Chloroplast Genomes

Physical mapping of the chloroplast genomes of *P. delavayi* var. *lutea*, *P. delavayi* var. *angustiloba*, and *P. ludlowii* was performed using OGDRAW (https://chlorobox.mpimp-golm.mpg.de/OGDraw.html; accessed on 3 November 2023) software [51]. The lengths and boundaries of the different sequences of LSC, SSC, and IR were compared, and information such as the size, GC content, and number of genes in the chloroplast genome of each species was obtained using Excel. Visualization of the chloroplast genomes of *P. delavayi* var. *lutea*, *P. delavayi* var. *angustiloba*, *P. ludlowii*, and their six related species was performed using IRscope (https://irscope.shinyapps.io/irapp/; accessed on 3 November 2023) [52], which was employed to compare the differences in the boundaries of the four regions of the chloroplast genomes of the different species.

### 4.4. Identification of Long Repeats and Simple Sequence Repeats (SSRs)

Repetitive sequences scattered among them were analyzed using the online software REPuter (http://bibiserv.cebitec.uni-bielefeld.-de/reputer; accessed on 6 November 2023), which includes four components: forward repeats, reverse repeats, palindromic repeats, and complementary repeats. The parameter settings were as follows: the minimum repeat length was set to 30 bp, and the similarity rate between two repeats was greater than 90% [53]. The simple sequence repeats (SSR) were identified by the software MISA (https://webblast.ipk-gatersleben.de/misa/; accessed on 6 November 2023), and mononucleotide, dinucleotide, trinucleotide, tetranucleotide, pentanucleotide, and hexanucleotide sequences were identified, with the minimum number of repeat thresholds set to 10, 6, 5, 3, 3 and 3, respectively. The minimum distance between two SSRs was 100 bp [54].

### 4.5. Codon Usage Bias Analysis

The cpDNAs of *P. delavayi* var. *lutea*, *P. delavayi* var. *angustiloba*, and *P. ludlowii* were excluded from sequences less than 300 bp in length and repetitive sequences. CDSs with ATG as the start codon and TAA, TAG, and TGA as the termination codon were extracted and combined to construct a fasta file. The relative synonymous codon usage (RSCU) was analyzed using Codon W1.4.4 software [55]. When the RSCU is greater than 1, the relative usage of the codon is high, and vice versa.

### 4.6. Chloroplast Genome Comparison

Using *P. delavayi* as the reference genome, the online software mVISTA (https://genome.lbl.gov/vista/mvista/about.shtml; accessed on 8 November 2023) was used to compare the chloroplast genome sequences of *P. delavayi* var. *lutea*, *P. delavayi* var. *angustiloba*, and *P. ludlowii*, and the shuffle-LAGAN model was used to detect the variation [56]. Then, DnaSP was used to calculate the nucleic acid polymorphisms, and sliding window analysis was performed, with the window length set to 600 and the step size set to 200 [57].

### 4.7. Selection Pressure Analysis

Protein sequences and coding sequences were extracted from *P. delavayi* var. *lutea*, *P. delavayi* var. *angustiloba*, and *P. ludlowii*. The homologous protein sequences were obtained by comparing them with the reference protein sequences using BLASTN (2.10.1) to find the best match. The homologous protein sequence was subsequently aligned using MAFFT (v7.427), and the aligned coding sequence was obtained by mapping the aligned protein sequence to the coding sequence using a Perl script. Finally, the ka and ks values were calculated using KaKs_Calculator2 [58].

### 4.8. Phylogenetic Analysis

To assess the phylogenetic relationships of Paeoniaceae, we established three datasets. The first dataset included 99 whole chloroplast genomes from Saxifragales. The second dataset included 20 chloroplast genomes from 18 Paeoniaceae species and 2 outgroup species. The third dataset included 73 protein-coding genes from chloroplast genomes of 18 Paeoniaceae species and 2 outgroup species. The unsequenced chloroplast genome in this paper was obtained from the NCBI GenBank (Appendix A). Based on the first datasets, the alinement was performed with MAFFT [59] and sheared by TrimAL [60]. ModelFinder was subsequently used to select the optimal model [61], and the chosen optimal model was GTR + F + R4. Finally, the phylogenetic tree was constructed by the maximum likelihood (ML) method, and the bootstrap value was set to 1000 [62]. Then, based on the second and third datasets, phylogenetic trees were constructed using the maximum likelihood (ML) and Bayesian inference (BI) methods, and the alignment process was consistent with that described above. The ML tree was constructed using IQtree, and the optimal model was chosen as K3Pu + F + R2, with a bootstrap value of 1000. The BI tree was generated using MrByaes, and the optimal model was GTR + F + I + G4. The trees were built using 4 Markov chains, the sampling frequency was set to 100, two mutually independent runs were performed, the initial 25% was removed as burn-in, 2,000,000 generations were run, and finally, a BI tree was used as a presentation with the posterior probability and the self-expansion value labeled at the nodes [63].

### 4.9. Divergence Time Dating

The time of divergence within the Paeoniaceae was estimated using BEAST v2.6.2 software [64,65]. Since there are no suitable fossils for Paeoniaceae, the divergence time estimates for Paeoniaceae based on fossils from several families by Li et al. [66] and Zhou et al. [67] were used as correction points. The age of the crown group of Paeoniaceae was approximately 28.00 Ma, that of *P. delavayi* and *P. lactiflora* was approximately 6.02 Ma, and that of *P. lactiflora* and *P. obovata* was approximately 2.83 Ma. The GTR model was employed as the site model. The log files were imported into Tracer v1.7.2 to compare the applicable molecular clock models. When an uncorrelated relaxed lognormal clock was used, the coefficientOfVariation was close to 0, indicating that the strict molecular clock model should be used. The Yule model was employed for the a priori value calculation of the tree for 10 million generations. The stability of the results was assessed after the run using Tracer v1.7.2 [68]; it was determined that the ESS values were all >200, and the results converged. The tree file was assembled using TreeAnnotator, resulting in MCC trees containing species divergence times, which were visualized using tvBOT [69].

## 5. Conclusions

In this study, we sequenced and assembled the chloroplast genomes of *P. delavayi* var. *lutea*, *P. delavayi* var. *angustiloba*, and *P. ludlowii* and comparatively analyzed the structures of and differences in their chloroplast genomes for the first time. These genes were highly conserved, containing 81–87 protein-coding genes, with both tRNA and rRNA numbers of 37 and 8, respectively. There were three types of long repeats and four SSR types, with the highest proportion of mononucleotide repeats occurring in SSRs, and most of them had A and T as the basic repeating units. Codon preference analysis revealed similar RSCU values, with a preference for ending at the A and U bases. Five regions with a high degree of variation were identified, namely, *psbK*-*psbL*, *petN*-*psbM*, *psbC*, *rps8*-*rpl14*, and *ycf1*, which can be used as vital barcodes for taxonomic studies. Selection pressure analysis revealed that the ribosomal protein small subunit gene rps18, associated with self-replication, is under positive selection and has rapidly evolved recently. Phylogenetic analysis clarified the phylogenetic relationships of Paeoniaceae and revealed that *P. delavayi* var. *lutea*, *P. delavayi* var. *angustiloba*, and *P. ludlowii* are monophyletic groups and sisters to *P. delavayi*. The estimation of divergence time revealed two evolutionary divergences of Paeoniaceae species in the early Oligocene and Miocene. Afterward, they underwent rapid adaptive radiation from the Pliocene to the early Pleistocene, when *P. delavayi* var. *lutea*, *P. delavayi* var. *angustiloba*, and *P. ludlowii* formed.

## Figures and Tables

**Figure 1 ijms-25-00257-f001:**
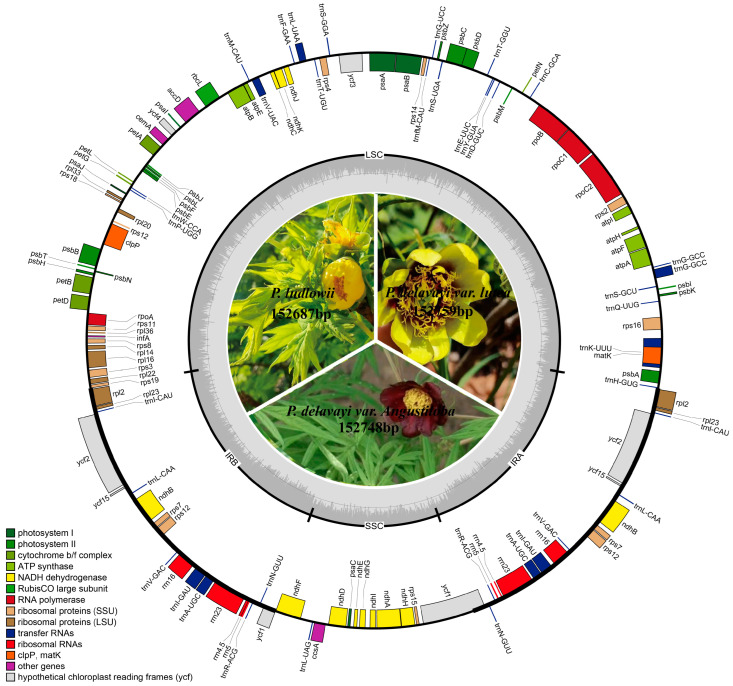
Chloroplast genome map of *P. delavayi* var. *lutea*, *P. delavayi* var. *angustiloba*, and *P. ludlowii*. The genes depicted inside the circle are transcribed clockwise, while the genes shown on the outside of the circle are transcribed counterclockwise. Different colors of genes that represent various functions are shown in the left corner of the bottom panel. The darker gray in the inner circle corresponds to the GC content, whereas the lighter gray corresponds to the AT content.

**Figure 2 ijms-25-00257-f002:**
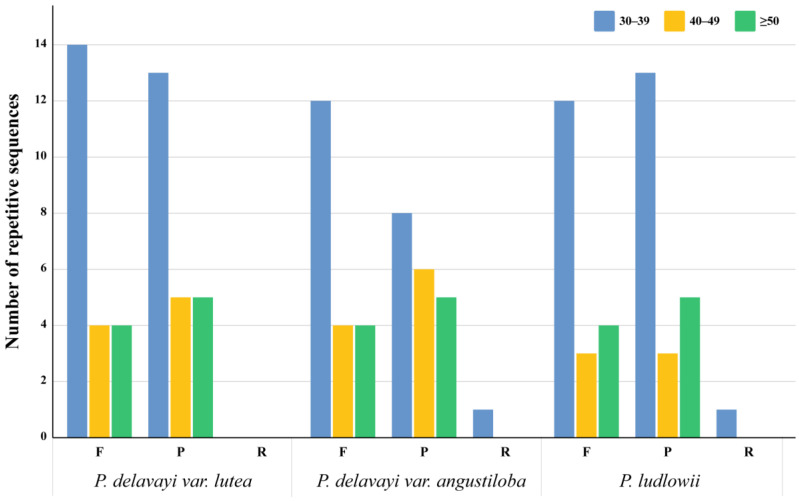
The number of long repeat sequences in the chloroplast genomes of *P. delavayi* var. *lutea*, *P. delavayi* var. *Angustiloba*, and *P. ludlowii* (F: forward repeats; R: reverse repeats; P: palindromic repeats; C: complementary repeats).

**Figure 3 ijms-25-00257-f003:**
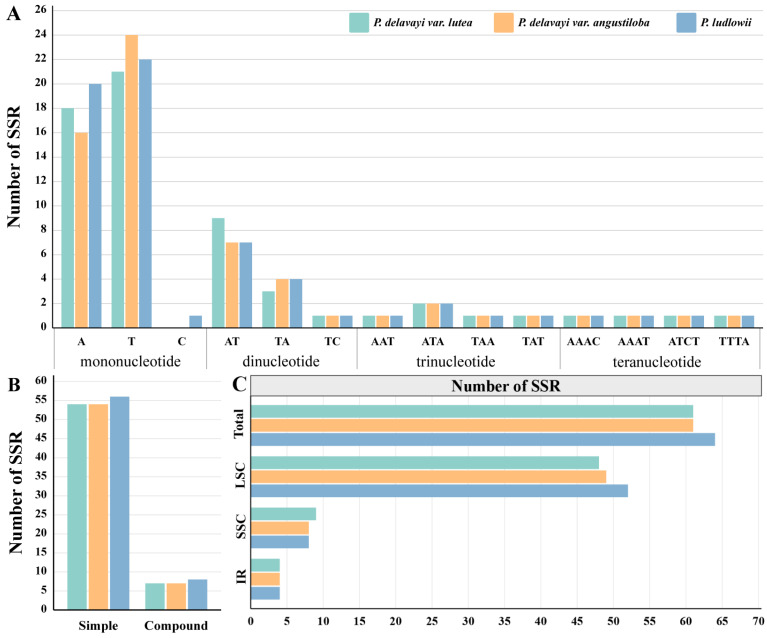
Analysis of simple sequence repeats (SSRs) in the chloroplast genomes of *P. delavayi* var. *lutea*, *P. delavayi* var. *angustiloba*, and *P. ludlowii*. (**A**) Number of different types of SSRs identified in the cp genomes. (**B**) Number of SSR repeats in regular and compound formations of the cp genomes. (**C**) SSR distributions in the LSC, SSC, and IR regions.

**Figure 4 ijms-25-00257-f004:**
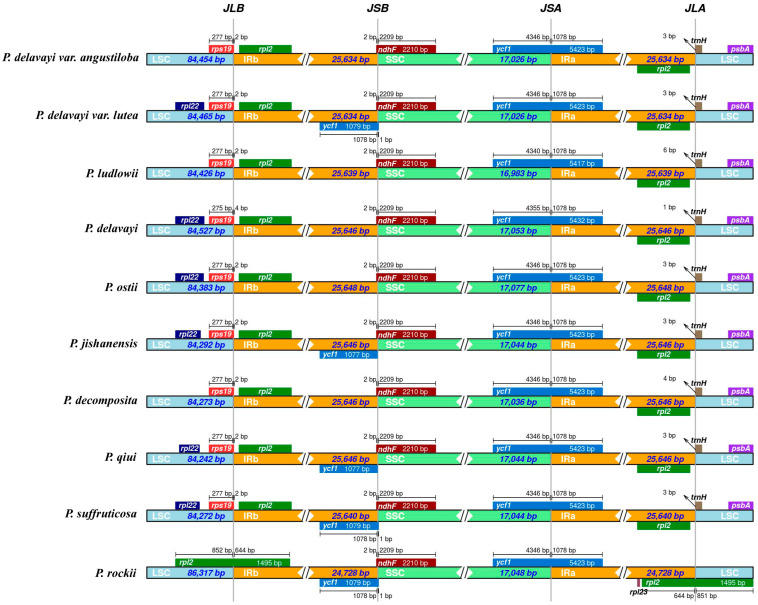
Comparison of the borders of large single-copy (LSC), small single-copy (SSC), and inverted repeat (IR) regions in the chloroplast genomes of ten Paeoniaceae species. JLB: junction of LSC and Irb; JSB: junction of SSC and Irb; JSA: junction of SSC and Ira; JLC: junction of LSC and Ira.

**Figure 5 ijms-25-00257-f005:**
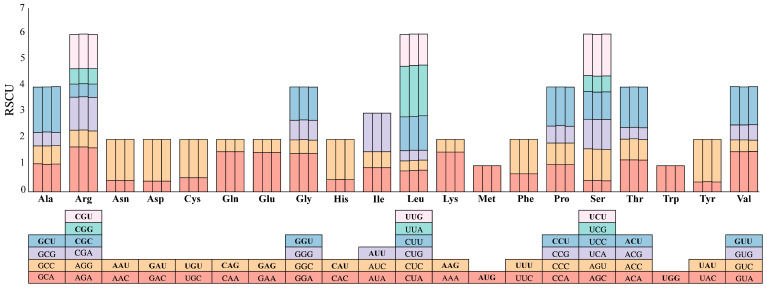
Relative synonymous codon usage (RSCU) in the chloroplast genomes of 3 Paeoniaceae species and the amino acids encoded by these codons. The order of the three columns is *P. delavayi* var. *lutea*, *P. delavayi* var. *angustiloba*, and *P. ludlowii*.

**Figure 6 ijms-25-00257-f006:**
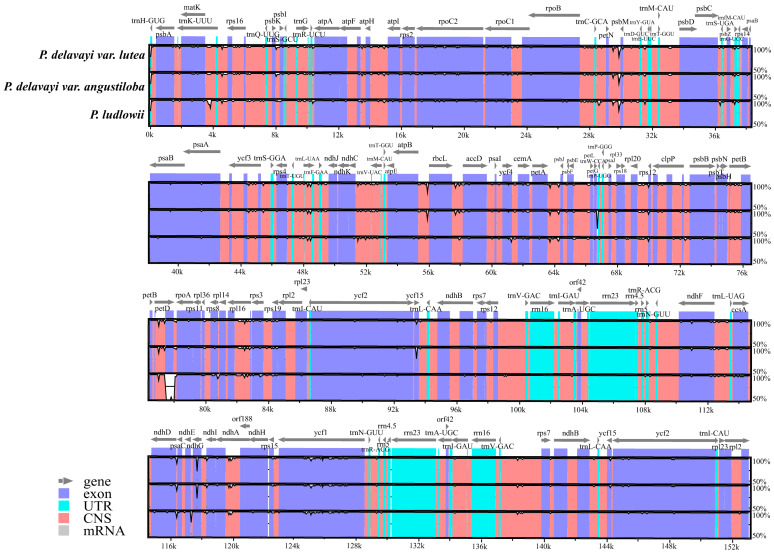
Alignment of chloroplast genome sequences in *P. delavayi* var. *lutea*, *P. delavayi* var. *angustiloba*, and *P. ludlowii*.

**Figure 7 ijms-25-00257-f007:**
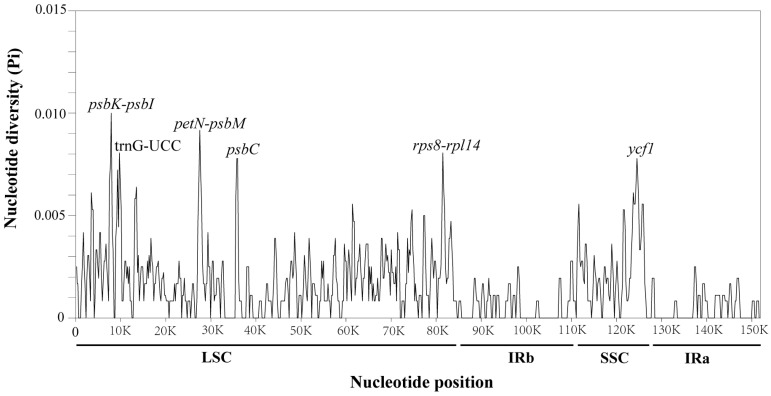
Sliding window test of nucleotide diversity (Pi) in *P. delavayi* var. *lutea*, *P. delavayi* var. *angustiloba*, and *P. ludlowii* (window length: 600 bp; step size: 200 bp). The *X*-axis represents the position of the midpoint of a window, while the *Y*-axis represents the Pi value of each window.

**Figure 8 ijms-25-00257-f008:**
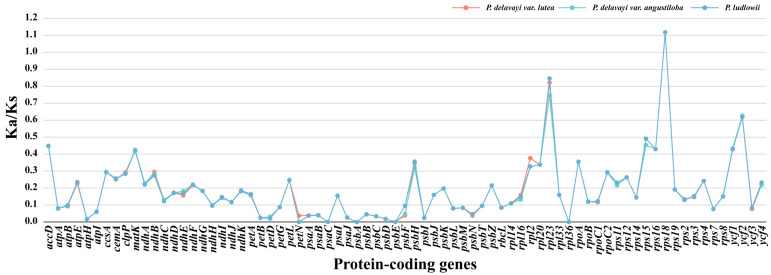
The Ka/Ks ratios of protein-coding genes in the chloroplast genomes of *P. delavayi* var. *lutea*, *P. delavayi* var. *angustiloba*, and *P. ludlowii*.

**Figure 9 ijms-25-00257-f009:**
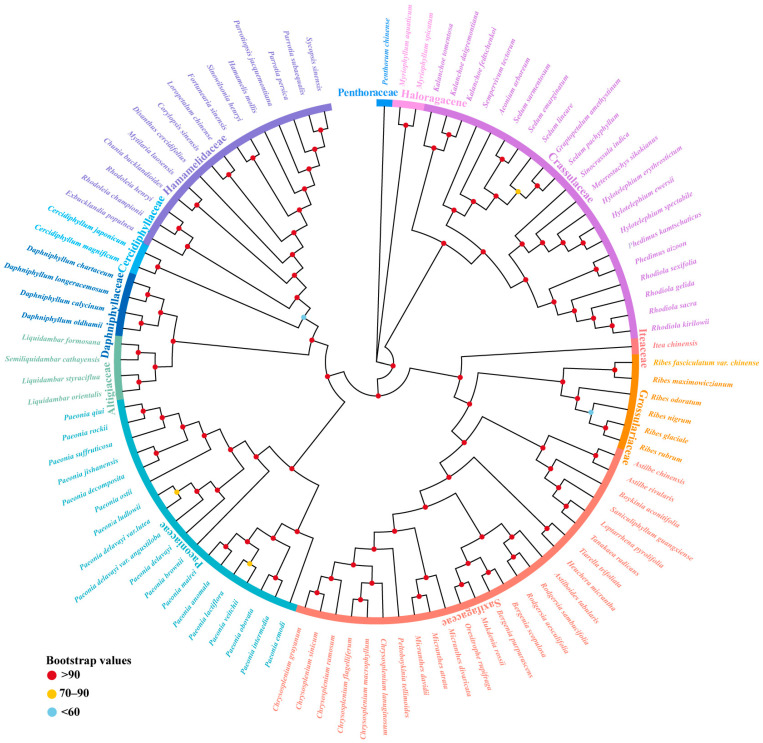
Phylogenetic tree based on WCG datasets of 99 Saxifragales species using ML methods.

**Figure 10 ijms-25-00257-f010:**
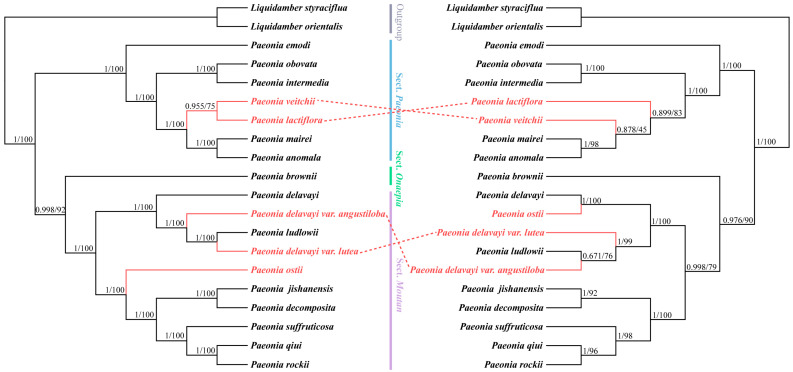
Phylogenetic reconstruction of Paeoniaceae using Bayesian inference (BI) and maximum likelihood (ML) methods based on the WCG (**left**) and PCG (**right**), respectively. The red branches represent the different topologies between the WCG and PCG trees. The values above the nodes are posterior probabilities (PPs) and bootstrap support (BS) values.

**Figure 11 ijms-25-00257-f011:**
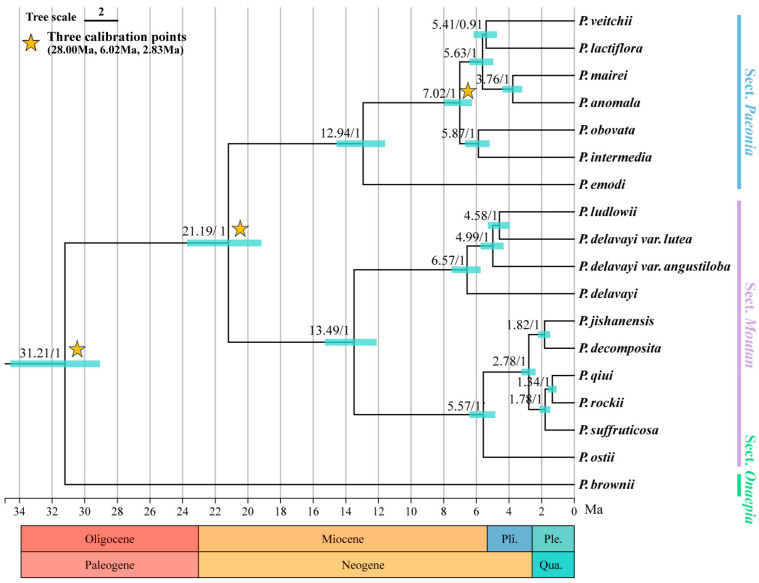
Divergence time estimation in Paeoniaceae based on whole chloroplast genomes with three calibration points (pentagrams). Branches in bold indicate the 95% HPD boundary. The horizontal coordinate is time (Ma).

**Table 1 ijms-25-00257-t001:** Comparison of the structural features of the chloroplast genomes of three species of Paeoniaceae.

Species	Genome Size (bp)	LSC (bp)	SSC (bp)	IR (bp)	Number of Genes	Protein-Coding Gene	tRNA	rRNA	CG (%)
*P. ludlowii*	152,687	84,426	16,983	25,639	128	83	37	8	38.44
*P. delavayi* var. *lutea*	152,759	84,465	17,026	25,634	132	87	37	8	38.41
*P. delavayi* var. *angustiloba*	152,748	84,454	17,026	25,634	126	81	37	8	38.41

**Table 2 ijms-25-00257-t002:** Annotated genes and their classification in the chloroplast genomes of three *Paeoniaceae* species.

Category of Genes	Group of Genes	Gene Name
Self-replication	Ribosomal RNAs	*rrn4.5* (×2), *rrn5* (×2), *rrn16* (×2), *rrn23* (×2)
	Transfer RNAs	*trnI*-*CAU* (×2), *trnL*-*CAA* (×2), *trnV*-*GAC* (×2), *trnI*-*GAU* (×2) ^d^, *trnA*-*UGC* (×2) ^d^, *trnR*-*ACG* (×2), *trnN*-*GUU* (×2), *trnL*-*UAG*, *trnP*-*UGG*, *trnW*-*CCA*, *trnM*-*CAU*, *trnV*-*UAC* ^d^, *trnF*-*GAA*, *trnL*-*UAA* ^d^, *trnT*-*UGU*, *trnS*-*GGA*, *trnM*-*CAU*, *trnG*-*GCC* (×2) ^d^, *trnG*-*UCC*, *trnS*-*UGA*, *trnT*-*GGU*, *trnE*-*UUC*, *trnY*-*GUA*, *trnD*-*GUC*, *trnC*-*GCA*, *trnR*-*UCU* ^a^, *trnS*-*GCU*, *trnQ*-*UUG*, *trnK*-*UUU* ^d^, *trnH*-*GUG*
	Ribosomal protein (small subunit)	*rps11*, *rps12* (×2), *rps14*, *rps15*, *rps16* ^d^, *rps18*, *rps19*, *rps2*, *rps3*, *rps4*, *rps7* (×2), *rps8*
	Ribosomal protein (large subunit)	*rpl14*, *rpl16* ^d^, *rpl2* (×2) ^d^, *rpl20*, *rpl22* ^a, b^, *rpl23* (×2), *rpl33*, *rpl36*
	DNA-dependent RNA polymerase	*rpoA*, *rpoB*, *rpoC1* ^d^, *rpoC2*
Photosynthesis	Subunits of photosystem	*psaA*, *psaB*, *psaC*, *psaI*, *psaJ*
	Subunits of photosystem	*psbA*, *psbB*, *psbC*, *psbD* ^d^, *psbE*, *psbF*, *psbI*, *psbJ*, *psbK*, *psbL* ^c^, *psbM*, *psbN*, *psbT*, *psbZ*
	Subunits of cytochrome b/f complex	*petA*, *petB* ^d^, *petD*, *petG*, *petL*, *petN*
	Subunits of ATP synthase	*atpA*, *atpB*, *atpE*, *atpF* ^d^, *atpH*, *atpI*
	Subunits of NADH dehydrogenase	*ndhA* ^d^, *ndhB* (×2) ^d^, *ndhC*, *ndhD*, *ndhE*, *ndhF*, *ndhG*, *ndhH*, *ndhI*, *ndhJ*, *ndhK*
	ATP-dependent protease subunit	*clpP* ^e^
	Subunit of rubisco	*rbcL*
Other genes	Maturase	*matK* ^b^
	Subunit of acetyl-CoA-carboxylase	*accD*
	Envelop membrane protein	*cemA*
	C-type cytochrome biogenesis	*ccsA*
	Conserved open reading frames	*ycf1* (×2), *ycf2* (×2), *ycf3* ^e^, *ycf4*, *ycf15* (×2) ^b^

(×2) Gene with two copies; ^a^ Gene lost in *P. delavayi* var. *lutea*; ^b^ Gene lost in *P. delavayi* var. *angustiloba*; ^c^ Gene lost in *P. ludlowii*; ^d^ Gene with one intron; ^e^ Gene with two introns.

## Data Availability

The chloroplast genome sequences of *P. delavayi* var. *lutea*, *P. delavayi* var. *angustiloba*, and *P. ludlowii* were deposited in the GenBank of the National Center for Biotechnology Information (NCBI) repository, and the accession numbers are OR187760, OR187761, and OR187762, respectively.

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
