# Peer review of "Complete Chloroplast Genomes and the Phylogenetic Analysis of Three Native Species of Paeoniaceae from the Sino-Himalayan Flora Subkingdom"

_ijms, 2023, doi:10.3390/ijms25010257_

Round 1
Reviewer 1 Report
Comments and Suggestions for Authors
The study sequenced and assembled the chloroplast genomes of P. delavayi var. lutea, P.delavayi var. angustiloba, and P. ludlowii, and conducted a comparative analysis of their structures and differences. The results revealed a high level of conservation, with 81-87 coding protein genes, 37 tRNAs, and 8 rRNAs. They possessed three types of long repeat sequences and four types of SSR, with single nucleotide repeats being the most abundant SSR type, and the majority of the basic repeat units being A and T. Codon preference analysis showed similar RSCU values, with a preference for A and U ending bases. Five highly variable regions were identified: psbK-psbL, petN-psbM, psbC, rps8-rpl14, and ycf1, which serve as important barcodes for taxonomic studies. Selection pressure analysis revealed that the ribosomal subunit gene rps18, which is related to self-replication, is under positive selection and has recently evolved rapidly. Phylogenetic analysis clarified the phylogenetic relationships of Paeoniaceae and revealed that P. delavayi var. lutea, P. delavayi var. angustiloba, and P. ludlowii form a monophyletic group, with P. delavayi as the sister group. Divergence time estimation revealed two evolutionary divergences in the Early Oligocene and Miocene of Paeoniaceae species. Subsequently, they evolved from a rapid adaptive radiation in the early Pliocene to P. delavayi var. lutea, P. delavayi var. angustiloba, and P. ludlowii. This study has certain research value, but there are several issues that need attention, as follows:
Main issues:
1. In the methods, total DNA of plants was extracted and chloroplast reads were extracted using GetOrganelle software. In the case of low genome coverage(less than 0.5~1X), the results extracted by the software method are not reliable. Has this study performed experimental validation on specific regions or genes? How many data was extracted for reads? What is the coverage? These details are not reflected in the results and need to be supplemented.
2. There have been previous reports on the chloroplast genomes of Paeoniaceae species (Li, H., Guo, Q. & Zheng, W. Characterization of the complete chloroplast genomes of two sister species of Paeonia: genome structure and evolution. Conservation Genet Resour 10, 209–212 (2018). https://doi.org/10.1007/s12686-017-0800-7), (Wu, L., Nie, L., Wang, Q. et al. Comparative and phylogenetic analyses of the chloroplast genomes of species of Paeoniaceae. Sci Rep 11, 14643 (2021). https://doi.org/10.1038/s41598-021-94137-0). Have the authors compared the completeness and differences between the chloroplast genomes of the same species and the ones assembled in this study? Which version of the chloroplast genome is currently the most complete?
Other issues:
1. The color combinations in the figures are inappropriate. A more distinct color should be chosen.
For Figure 1, the Latin names of the three species can be in black.
For Figure 9, the Latin names of the species are not clear and the colors are too light. It is recommended to use a different color scheme.
The colors of the three dots representing bootstrap values in Figure 9 are also recommended to be changed.
For Figure 10, bold lines, especially the dotted lines.
Comments on the Quality of English LanguageThe paper should be proofread by a professional English speaker to meet the requirements of the journal.
Reviewer 2 Report
Comments and Suggestions for Authors
This study described the characteristics of three newly sequenced chloroplast genomes of P. delavayi var. angustiloba, P. delavayi var. lutea and P. ludlowii in the family Paeoniaceae. Comparison among the three assembled chloroplast genomes showed few variations in the genome length, numbers of genes as well as in types of SSR regions. Using the three chloroplast genomes and several obtained data, the authors estimated the phylogenetic relationships and infra-section divergence events of Paeoniaceae, suggesting a monophyletic group of the three sequenced species.
The three chloroplast genomes were well assembled and analyzed, which provided useful genetic resources for Paeoniaceae. However, in my view the INTRODUCTION part was poorly organized, thus leaded to limited significance of the three chloroplast genome data on scientific issues, for example the phylogenetic relationships of species in Paeoniaceae. Moreover, this MS version should be carefully checked and improved as several mistakes were found. Other comments are listed below:
1, the INTRODUCTION part needed to be modified, and for most part of the MS English need to be largely improved.
2, line23: DNA molecular markers for?
3, line24: what kind of data were used for phylogenetic analysis?
4, line50-51: please check this sentence.
5, line106: AU content?
6, line115-116: “six tRNAs and ...” of which species?
7, line133: what is “chloroplast genome strips”?
8, line146: the highest content of?
9, line160-164: these descriptions can be moved to the introduction part.
10, line179-182: as the same to the above point, these sentences can be moved to the introduction part.
11, line211: that should be “mVISTA”?
12, line229: Ka/Ks?
13, line237: Saxifragalesis?
14, line300: this inference was overestimated that needed to be verified by the authors.
15, line310-311: please add related references for this sentence.
16, line319-320: also, this sentence required references.
17, line329-334: this part was redundant that had been stated in the result part.
18, line336, delete “those of”
19, line357-359: references were required
20, line386-387: this statement was obviously wrong because the inferred divergence time among sections of Paeoniaceae was dated in the early Miocene.
21, line491: for the BEAST work, a prior test on molecular clock models was suggested to select a suitable model for dating species divergence, however, the authors just used the strict clock model without model comparisons.
Figure 4, please explain “JLB, JSB, JSA and JLA”
Comments on the Quality of English LanguageFor most part of the MS English need to be improved
Reviewer 3 Report
Comments and Suggestions for Authors
In this study, are sequenced, assembled, and annotated the chloroplast genomes of P. delavayi var. lutea, P. delavayi var. angustiloba, and P. ludlowii. Their chloroplast genomes are comparatively analyzed, and their phylogenetic relationships and divergence times are inferred. Divergence time estimation is done and two evolutionary divergences are described as follows: in the early Oligocene and Miocene and later on a rapid adaptive radiation takes place in the early Pleistocene, when the P. delavayi var. lutea, P. delavayi var. angustiloba, and P. ludlowii are formed. The paper is well written and the results are convincing.
Comments
I think that some publications that are related to the topic should find their place in the discussion or in the introduction.
Ji, L., Wang, Q., da Silva, J. A. T., & Yu, X. N. (2012). The genetic diversity of Paeonia L. Scientia horticulturae, 143, 62-74.
Xiao, P. X., Li, Y., Lu, J., Zuo, H., Pingcuo, G., Ying, H., ... & Jiao, W. B. (2023). High-quality assembly and methylome of a Tibetan wild tree peony genome (Paeonia ludlowii) reveal the evolution of giant genome architecture. Horticulture Research, uhad241. https://academic.oup.com/hr/advance-article/doi/10.1093/hr/uhad241/7420485
Dong, L., Hong, M., Li, Z. H., Liu, X. X., & Zhang, Y. L. (2011). Karyotypic Studies of five Paeonia ludlowii populations from China. Caryologia, 64(4), 370-376.
De-yuan, H. (1997). Paeonia (Paeoniaceae) in Xizang (Tibet). Novon, 156-161.
Round 2
Reviewer 2 Report
Comments and Suggestions for Authors
Previous comments on the MS were all modified by authors in the revision, which can be accepted for publishment.